# Stormwater Utilities: A Sustainable Answer to Many Questions

**Carlos Novaes \* and Rui Marques** 

CERIS, Instituto Superior Técnico (IST), University of Lisbon, Av. Rovisco Pais, 1049-001 Lisbon, Portugal; rui.marques@tecnico.ulisboa.pt
\* Correspondence: cnovaes.augusto@gmail.com

**Abstract:** One of the most complex and difficult questions to answer concerns how to organize and economically support public services of all kinds. In terms of services that involve a multiplicity of actors and objectives, as is the case with urban stormwater management, the difficulty is magnified and resources never seem to be sufficient. This paper reviews the successful approaches to stormwater management in a number of countries and concludes that it is both feasible and possible to successfully structure stormwater management in cities using a variety of models and incentives. With examples from cases practiced in the USA and Canada, based theoretically on the user-pays principle and on the fair distribution of impacts, the text innovates showing not only a technically and legally viable option, but an opportunity for users to become aware of the importance of reducing environmental impacts. By raising the possibility of delivering services out of the general public budget, reducing the taxation of all in exchange for charging only users and improving the performance, the discussion is directed, in an innovative way, to a very rarely questioned aspect and links the change in mentality from and economic way of thinking towards the new stormwater paradigm shift and SDGs.

**Keywords:** drainage; fees; financing; stormwater management; sustainability; utilities

## 1. Introduction

The management of urban stormwater evolved with the understanding of its role and the good and bad characteristics that such water brings to society. In each era, water has had different roles, but has always been connected with human activities, and cities were born and developed in close relationship with water. The close presence of water, whether for agriculture, energy production, navigation or consumption, in most cases brought value to the territories, with the exception of urban flooding [1].

Cities continued to grow and man thought wastewater and stormwater were reasons for diseases, and that it was necessary to take it as soon as possible to rivers, lakes, estuaries and the sea. The function of the receiving water bodies was not only transport, but effluent discharge, without treatment, as is still the case in many places today. In the era of 'tout à l'égout' [2], the hygienist phase of the 19th century, the reigning order indicated was to move waste and rainwater away from the cities as quickly as possible, not only to avoid the proliferation of diseases, but also to prevent flooding. The concern was fundamentally hydraulic, with a focus on quantities and mass flows, i.e., the problem was reduced to the sizing of pipes and channels, a matter for engineers. Concerning stormwater drainage, there was an understanding that these were of good quality and free of pollutants [3], reinforcing the idea that only volumes, mass flows, and dilution should be addressed. As for quality, represented by pollution, people said that it was only a matter of diluting polluted volumes into larger volumes, free of pollutants. The issue of sizing structures for drainage and dilution went through the discovery of which rains could be predictable in each region, a problem of hydrology and statistics, the latter helping by informing the probability of occurrence of certain volumes and the associated risks, expressed by determining the so-called return times of the events. Again, problems and issues are more related to engineers. From the possession of this information, the problem of the decision to

choose return periods, or "project rainfall", depended on political decisions, closely linked to investments to be made in physical infrastructure and their respective risks.

With the advances brought by microbiology and epidemiology and Koch's discoveries regarding cholera, the understanding grew that the issue of wastewater quality was of crucial importance allowing sanitary approaches to sanitation and drainage to occupy a prominent place; it was the beginning of the sanitarian phase after the hygienist phase of the 19th century. The first wastewater treatment plants started to be developed at the turn of the century [4]. The decision about the type of treatment and size of the facilities involved the determination of the desired quality levels for effluents, an issue that no longer affected only engineers, but also health professionals, considering the risks that society would be willing to undergo and the alternatives of investments to be made, i.e., again, decisions of a political nature.

Currently, as of the nineties, there is an upsurge in urban flooding problems for several reasons, with two of them more relevant and interconnected: demographic growth, with the consequent territorial expansion of the urban fabric; and climate change, leading to the perception that just draining downstream, ever further and with greater volumes, treating rainwater effluents in a concentrated manner, at the point of discharge ("end of pipe"), in a vision on the one hand hydraulic and on the other hygienist, has not even found physical spaces for the task [5].

Later on, simultaneously with the environmental and right to the city movements, the vision of sustainability took over urban environments that started to understand water not as a problem anymore, but as a solution to old and new issues of quantity and quality, such as scarcity, well-being and comfort, exemplified, respectively, by its use in daily life, embellishment and the fight against "heat islands" [6].

According to this approach, alternative techniques emerge, in opposition to the traditional method of removal, which mainly uses buried pipes. The new mentality is one of harmonious coexistence with water; therefore, in addition to its visible presence on the surface, there must be treatment at the source, that is, as close as possible to the places of origin, where precipitation occurs [7].

In terms of management, what worked before—the centralized management with command and control concentrated in the public power—requires changes, and a change in the traditional management paradigm. With new and multiple actors assuming different roles, disputing and sharing the resource represented by urban stormwater, practices are gradually changing and favoring the decentralization and democratization of decisions [8].

At the same time, the change reaches aspects of service funding, as the understanding had always been that flooding issues, linked to large volumes and flows (hydraulics), were usually borne, at great expense, by general (centralized) public budgets. The feasibility of solutions at the source, or located where precipitation occurs, generally requires potentially lower and decentralized expenditures enabling private sharing and participation in the solution in a distributed manner [9].

Around the world, this paradigm shift is being studied to create the best format for financing and management. The USA and Canada, countries where there have been successful experiences for some time (around three decades [10]), materialized through stormwater programs, are partially or totally based on the collection of tariffs from users. Users became important and stable sources of funds, specifically for the improvement in urban stormwater management systems to comply with legislation. In the USA and Canada, the collection mechanisms for funding the activity are called stormwater utilities (SWUs), with specific characteristics in each place, but which deserve to be observed in order to learn from them.

The contributions of this paper lie in the approach to a subject absent from the literature, despite its importance, exemplified by the number of existing cases and of importance of the countries in which it is presented. Despite this, however, SWUs are financing stormwater management mechanisms that are little used in most countries where funding is required. In this sense, the contribution is to expand the dissemination and discussions around the

subject and its application, so that through its knowledge its use can be expanded and the improvement in stormwater management can be achieved.

Besides this short introduction, the article is structured in four more sections. The Section 2 will focus on issues related to the origin and reason for the emergence of SWUs. In the Section 3, several cases where SWUs exist are presented; in the Section 4 the results and discussions are briefly presented; finally, in Section 5, the conclusions are drawn. Figure 1 presents a flow chart of the study with each chapter, its main aspects, as well as some of its interconnections.

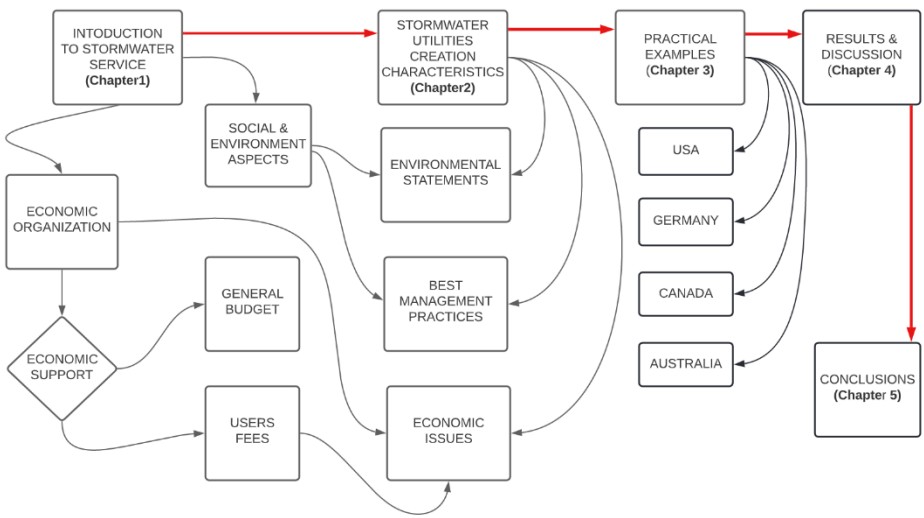

**Figure 1.** Study flow chart.

## 2. The Creation of SWUs

In line with the diversity of aspects to be addressed, new demands for resources and management have arisen from the entry into force of legislation and regulations targeted at improving the quality of urban service delivery systems, including stormwater management. Costs expected to result from climate change (increased frequency, intensity, and duration of rainfall with flooding and rising sea levels) have led to a search for alternatives to system funding through the application of the user-pays principle, based on charges levied on users according to their contribution to runoff, supporting, even if sometimes only partially, the new costs to be assimilated by budgets [11]. In each location, these financing methods, often associated with new management practices, have become institutionalized and have been given different names: SWU, stormwater fee, stormwater user fee, and stormwater service fee [12].

Cities support increasing costs, but with no proportionally crescent budgets, of maintenance and replacement of ageing infrastructure, of the new built areas and of quality and quantity costs occurring depending on climate change. In Canada, due to climate change, the stormwater infrastructure is considered in critical condition, but it was built in the last twenty years. With an estimated lifetime of 70–100 years for linear systems, 50–80 years for structures, and 25 years for electrical and mechanical components, this picture is surprising.

The SWUs, mechanisms to obtain resources for financing urban stormwater management, resulted from the perception by the American municipalities of the need to find economic means to cope with the increasing pressure on their budgets to meet socio-economic and environmental demands. These demands were incorporated into the American legislation through water quality control requirements due to diffuse urban stormwater pollution, considered a pollution point source at the point of discharge.

The legal and regulatory framework was developed over the last five decades since 1972, with the Clean Water Act (CWA) aimed at controlling water quality, revised in 1987 through the Water Quality Act (WQA), with provisions added for five categories of stormwater discharge. These provisions, classified under Phase I, established a set of

restrictions for large and medium-sized municipal storm sewer separator systems (MS4). These systems cover populations greater than 100,000 and according to discharge quality permits, bring them into compliance with the National Pollutant Discharge Elimination System (NPDES). In 1990, the final rules were established for Phase I, and later, in 1999, for Phase II. In small MS4, in addition to systems for fewer than 100,000 inhabitants, there are industries and construction areas of 4047 to 20,234 m$^2$ (1–5 acres). Stormwater eventually carries pollutants such as nutrients, pathogens, sediment, and metals, but must fall within the limits of the Total Maximum Daily Load (TMDL), a pollutant load that can be discharged to a given receiving body, without failing to meet the quality standards established by the states. The TMDL program applies to all MS4 systems, industrial and construction activities, and its limits include both point source and diffuse source loads [13].

Additionally, from the 1980s onwards, partly as a reflection of the taxpayer revolts of the 1970s, and the passage, in 1978, of proposition 13 in California, which placed limits on property taxation, some governments began to consider tariffs as a better source of resources than taxation for urban services. Thus, more favourable conditions were created for the introduction of stormwater utilities, a period considered as the utility model different from the traditional tax-supported public works model, until then predominant [14].

The creation of SWUs in the USA is not mandatory and depends on the perception of their need by the populations, policymakers, and those in charge of state legislations, on which SWUs depend to be implemented. There is no pre-defined size, with small communities such as Indian Creek Village in Florida (only 88 people), according to the 2010 Census population, and Los Angeles (over 4 million inhabitants).

There are nine models for calculating collection rates: dual, flat, tier, square foot, parcel acre, meter, usage, equivalent residential unit (ERU), and residential equivalent factor (REF). Some are based on impervious areas, such as the most widely used, ERU. Others are based on runoff generated, such as REF, or other forms of measurement. ERU is more popular in places with high population density and high property values while flat fee is more popular in places with low population densities and low property values [12].

Thus, in the absence of a general rule, the junction of necessity and opportunity has favored the creation of SWUs. The opportunity often occurred after catastrophic events, such as the hurricanes Katrina, in Louisiana (LA) and Mississippi (MS) fifteen years ago, and Sandy, in Connecticut (CT), New Jersey (NJ), and New York (NY) eight years ago, after which, however, by 2021, there were still no SWUs in place.

In the USA, the coming into force of legislation showed a relationship with the higher number of deployed SWUs [12], as also a clear definition of the legal authority in charge of each city, county, and watershed. In the some way, important was the key role played by the professional organizations, providing information, support, and encouragement to communities interested in the implementation of SWUs.

## 3. Stormwater Utilities (SWUs)

### 3.1. The USA

In the US, according to research by the University of Kentucky, there are 1851 distributed SWUs identified, in 41 states and the District of Columbia [15]. While the number of SWUs may seem large, it becomes small when compared to the number of 22,389 communities computed as participants in the National Flood Insurance Program (NFIP), as of June 2019, meaning there are SWUs in less than 10% of this total. Despite a long history of SWU implementation in the US, the main challenges that remain for communities, regardless of their size, are related to adequacy of funding and public support, which are compounded by aging infrastructure [16].

The national average monthly fee paid by single-family homes is USD 5.94, increasing over time according to the consumer price index (CPI), with values varying from USD 0 to 45, although there are situations where reductions may occur and the range may reflect stormwater needs and also political contexts. The most widely used calculation method is based on the impervious areas of the land and on the ERU system, an average of the

single-family residential impervious areas of land, but in some communities, a value can be defined based on the average of all the areas of residential land. The method calculates the amount to be charged based on the impervious areas of the lots, regardless of the total areas. The ERU is calculated through sampling carried out through field research [17], but can also be estimated through aerial or satellite images. Once the total impervious area of residential properties (AI) is obtained, it is divided by the number of properties, giving the ERU value [15].

For non-residential land, the rates are proportional to the ratio of the impervious area of the land to the ERU. The most commonly found average size for ERUs (895 utilities) in the University of Kentucky survey was 3072 square feet of impervious area, so it is important to determine ERUs accurately so that no one pays a disproportionate amount. There are other taxation systems, such as tiered systems (254 utilities) or flat fees (230 utilities). The ERU can be considered a system of infinite levels or steps and the flat fee and dual fee (108 utilities) as systems of a single level or step, the latter considering taxation for residences and another one for non-residential properties.

An example, taken from research conducted in 2021 by the University of Kentucky (Campbell, and Bradshaw 2021) [16], illustrates calculation systematics for a hypothetical area and allows conclusions to be drawn: residential waterproofed area = $15 \times 10^7$ sq ft; non-residential waterproofed area = $15 \times 10^7$ sq ft; ascertained ERU = 3000 sq ft; annual amount required for the selected level of service = CD\$ 12 million and every household pays a fee of 1ERU.

Dividing the total waterproofed area ($30 \times 10^7$ sq ft) by the standard computed ERU gives a number of 100,000 ERUs, 50% of which are residential areas and 50% of which are non-residential areas; therefore, a monthly fee of CD\$ 1 million is required, which when divided by the number of 100,000 ERUs indicates a base amount of CD\$ 10 per ERU per month.

If, however, for example for political reasons, it is decided that the assessed value to be used for non-residential areas should be for standard ERUs with 4000 sq ft and not 3000 sq ft, the number of non-residential ERUs becomes no longer 50,000 but 37,500 ($1.5 \times 10^7$ sq ft divided by 4000 sq ft), totaling no longer 100,000 but 87,500 ERUs in the municipality (50,000 residential and 3700 non-residential), which implies a value no longer of CD\$ 10/ERU, but of CD\$ 11.43 per ERU ($87,500 \times 11.43 =$ CD\$ 1 million per month) and, according to the following Equations (1) and (2), the percentage of costs will no longer be 50% between residential and non-residential areas, but 57% for residential and 43% for non-residential.

$$Frac_{res} = ERU_{res}/ERUs_{res} + IA_{nonres}/ERU_{used} \tag{1}$$

$$Frac_{nonres} = 1 - Frac_{res} \tag{2}$$

where $Frac_{res}$ corresponds to the fraction of the stormwater program paid by residential customers; $ERU_{res}$ is the total number of residential ERUs in the city; $IA_{nonres}$ is related to the total non-residential impervious areas in town; $ERU_{used}$ is the actual ERU used as opposed to the true ERU.

Similarly, if the standard ERU in the non-residential area with the value of 2000 sq ft is used, 125,000 ERUs will be obtained (50,000 residential and 75,000 non-residential), whereby the value of the ERU equals CD\$ 8 ($125,000 \times 8 = 1$ million) with the residential area bearing 40 and the non-residential area bearing 60% of the costs. Similarly, for ERU = 1000 sq ft, the percentages of monthly costs borne become 25% for residential areas and 75% for non-residential areas, or for ERU = 5000 sq ft, 62.5% for residential and 37.5% for non-residential users, respectively, according to Equations (1) and (2). From this, it is clear that the determination of the ERU is a very important aspect to have a fair taxation system that reduces the possibility of questioning.

Still, as to the example, it should be highlighted that it does not take into account possible reductions, applied in some municipalities, due to the placement of rainfall retention

devices on lots or even the disconnection from collective drainage systems, besides other aspects that motivate exemptions.

The second most popular model is the REF method, with 133 utility cases; this system is based on the amount of runoff from a unit compared to the amount of runoff by a standard property of a single-family dwelling, considering an event with a determined return time, for example, 2 years and 24 h of rain, calculated by the rational method or the Soil Conservation Service (SCS). Besides relying on hydrological information over time and soils, this system penalizes commercial properties for shorter return times and residential properties for large return times [18].

Thus, building a model for calculating fair taxation, making SWUs accepted by all as a development factor, is still a complex and evolving task that depends on several parameters in addition to policy options in each location and different development context. Nevertheless, some SWUs have made significant capital investments through user fee programs, such as in Fort Collins (CD$ 120 million), Bremerton (CD$ 55 million), and Raleigh (CD$ 100 million), initiated in 1980, 1994, and 2004, respectively, and in the second case the investment is to promote the separation of the existing unitary system [16].

In 2021, most of the 73 participants in the survey, which covered 20 American states, conducted by the consulting firm Black & Veatch, declared to: have a separating system (82%); have a municipality as their area of jurisdiction (97%); carry out the collection of drainage fees on water and sewerage bills (78%); consider a drainage website the most effective means of ensuring approval and support for the fees charged to users; and to fit into Phase II (population under 100,000) of the EPA's Municipal Separate Storm Sewer Systems (MS4s) discharge regulation program.

EPA has 855 participants in Phase I MS4s (population over 100,000) and 6695 in Phase II MS4s which include many cities and regions. In the majority, i.e., in 54% of those locations, where combined systems still exist, the combined systems account for less than 25% of the total system. Despite being the third most important item cited in the survey and that 73% of systems have—according to the survey—aging drainage infrastructure, asset management plans are in place in only 63% of systems falling under MS4s Phase I and 35% falling under MS4s Phase II.

The main percentages of instruments used for funding corresponds to cash (78%) or debts (22%), according to the percentages of answers for each type of instrument. As for the main sources of revenue, 95% of the answers indicate that more than 75% of the amounts are supported by fees received from users and the three main activities described as included in the O&M budget are: illicit discharge detection and elimination (96%), best management practices (90-92%) and public education (92%) [16].

*3.2. Germany*

In Germany, in several cities since the 1990s, based on the polluter-pays principle, stormwater management charges have been introduced taking into account the impervious area. Since many cities have single systems, i.e., systems that deal with both stormwater and wastewater systems, they are charged jointly through a single fee and the calculation is based on water supply consumption, which is not a fair way of charging. From the idea of changing to a fairer system, based on the mentioned principle, in most states the fee for impervious areas was introduced, but just with a value around only 20–75% of the costs of stormwater and wastewater management [17].

There are two ways to calculate the impervious area: by estimation, according to zoning (Munich, since 1970), or by measurement (Hamburg, since 2012; Dresden, since 1998; and the State of Baden-Wuerttemberg, since 2010). Calculation by estimation is easier to implement but more inaccurate.

The implementation of the levy resulted in: waterproofing area reductions of 4.5 M m$^2$ or 240,000 m$^2$/year in Munich with 3000 ML groundwater recharge; 10% reduction in waterproofing area per person in Dresden; and in the state of Baden-Wuerttemberg, 48% of

the cities reported decreases, 11% with high reductions already in the first two years after the levy implementation.

In Munich, maps with colors identify the runoff coefficients, being 0.9 for the blue zone, in the city center; 0.6 for the pink strips, in intermediate regions, between the center and the outskirts; 0.5 for the outer suburban areas (orange); and 0.35 for residential plots in the outer suburbs (green) (Vietz et al. 2018). Additionally, several beneficial effects were noted, such as the reduction in the quantities treated in the combined systems in Munich and Dresden, enabling process optimization and deferring infrastructure upgrades of existing systems [17].

*3.3. Canada*

In Canada, only 4 out of 48 utilities use the ERU system and eight use property value or "ad valorem" taxation and the average taxation is CD$ 10.67 [15].

In the Victoria Community, located in British Columbia, integration between stormwater management (SWM) and street sweeping has recently been discussed with the latter usually being included in property-related fees [19].

In Victoria, stormwater bills are issued annually to property owners and are determined based on property-specific characteristics such as impervious areas (roofs, car parks, and driveways) measured with the aid of building plans, aerial photography and mapping using georeferencing technology (GIS).

The value, in 2022, is of CD$ 0.654 per square meter and street cleaning is determined by frontage length and street type, charged per meter of lot frontage, varying according to Table 1 below.

**Table 1.** Victoria charge, according to the type of street.

| Street Type | CD$/Meter of Street Frontage |
| --- | --- |
| Local streets | 1.81 |
| Collector streets | 3.84 |
| Arterial streets | 4.35 |
| Downtown streets | 43.60 |

The property impact on the stormwater system through a flat portion of the charge, based on the building code is: low density residential—CD$ 0; multi-family residential—CD$ 81.79; civic/institutional—CD$ 72.98, and commercial/industrial—CD$ 148.38. Finally, there is a program in which property can be registered if they have ten or more parking spaces and are self-businesses, recreational facilities, recycling operations, storage yards, or have construction activities on site, paying CD$ 169.70 per year (2022).

In a study conducted in the City of Thunder Bay, comparing the various forms of taxation for stormwater funding, the conclusions were that urban properties subsidize rural areas by approximately CD$ 300.00 annually and that residential properties account for 67% of the contributions to the stormwater program while non-residential properties receive the remaining 33%.

However, the runoff from residential areas is only 58% and the remaining 42% comes from non-residential areas, meaning that a distribution based on contributing areas would lead to a 9% redistribution, i.e., the average residential properties would bear 9% less while the non-residential areas would bear an average of 9% more in levies.

Although this distribution would be fairer, one of the recommendations of the study was that the change would only be worthwhile above 10%, given the high implementation costs of reallocating only CD$ 360,000 per year from residential to non-residential plots, against a budget of CD$ 4 M and minimal apportionment of the difference among the 38,203 existing residential properties.

There are, however, other aspects that should be considered in the long term, such as the observation that property tax encourages urban sprawl while the user fee option encourages densification, and other issues including environmental liabilities [20].

### 3.4. Stormwater Management in Australia

The changes that occurred in water management in Australia over the past five decades determined its current state, according to a path dependence viewpoint (Otoch et al. 2019) [21].

However, according to Brown and other scholars, moments of tension and alignment existed between six distinct institutional logics (decision making, risk, sustainability, water quality, infrastructure, and demand) that coexisted in permanent evolution. Thus, according to a study conducted in the period between 1970 and 2015, throughout all that time, the urban water management sector in Australia showed great complexity [22].

In this context, the evolution (rather than a revolution) towards the current practices of Sustainable Urban Water Management (SUWM), which emerged in the 1990s, was based on the trajectory traveled by the logic of sustainability, water quality, and demand.

Changes in the idea of sustainability were characterized by the focus on aquatic health and the reorientation of the vision of point source pollution to diffuse, reflected in the logic of water quality that has become more restrictive with standards and monitoring based on aquatic ecology.

The demand was characterized by the emancipation of the end-user and the growth of expectations related to urban amenities and environmental protection. The infrastructure model, identified with engineering expertise, evolved from civil engineering based on forecasting and control to a more significant multidisciplinary approach, in which adaptability and flexibility became important design parameters.

The decision-making logic also evolved from the seventies, when it was characterized by bureaucratic, paternalistic decisions, centralized in powerful, vertically integrated organizations, and focused only on water supply, treatment, and drainage.

As of the 1980s, the government's influence grew, with the private sector participating and the predominance of the economic efficiency viewpoint. Decisions considered economic factors first, causing the user-pays principle to take the place of the tenure principle, which had property as the determining value in pricing decisions.

As of the 1990s, this vision intensified, reflecting free-market competition, with a commercial focus and financial instruments in decision-making. Public–private partnerships, for instance, are considered an alternative, and the user acquires the status of a consumer.

This framework remains from the 2000s with the water markets [23] when, however, due to the "millennium drought" at "a critical juncture", according to the definition of historical institutionalism [24], the logic of risk comes into play, temporarily interrupting decentralization, and the construction of large centralized desalination structures in all major cities [22].

Australia has a federative system of government, commonly referred to as "Commonwealth" or federal, with six states, two territories, a constitution, proclaimed in 1901, which defines the roles of each of the eight federal entities and, according to section 100, water management is the responsibility of each.

In the states and territories, there is another level of local government, which are the municipalities and district councils. In most states, the state governments own the utilities and local governments do the planning and management of stormwater services and the systems are separative.

For over thirty years, Australia has been developing its national water quality management strategy. It includes the use of stormwater for supply and guidelines are available for adoption by the states and territories. There are also guidelines for the evaluation of Water-Sensitive Urban Design (WSUD) options that incorporate an integrated approach to the urban water cycle. This includes the management of water supply, sewerage, groundwater, stormwater, land use, and environmental protection.

### 3.4.1. The Salisbury Example—South Australia

The city of Salisbury, in the metropolitan area of Adelaide (population 1.3 M), South Australia, developed through rapid urbanization from the 1970s onwards. Today, with around 137,000 inhabitants and an average annual rainfall of around 430 mm, mostly occurring in winter, it adopted WSUD principles to maximize the use of run-off water and reduce the risk of flooding. Aquifer recharge management was introduced to control the low salinity of stormwater by using Aquifer Storage and Recovery (ASR) in a brackish aquifer for subsequent irrigation.

The stormwater is collected in retention basins forming wetlands and lakes and subsequently infiltrated into the aquifers, with retention time around seven to ten days, being recovered through ASRs or Aquifer Storage Transfer and Recovery (ASTRs) allowing the reduction in the demand for water supply used for irrigation of sports fields.

Wetlands now occupy about 200 ha of the catchment area and in 2001 the City of Salisbury expanded the use of urban stormwater as a commercial enterprise through a public–private partnership project. The project focuses on applying AUD 4.5 million to construct wetlands and ASR facilities for stormwater treatment and storage, at Parafield Airport, a secondary airport in Adelaide. In this case, also a purple pipe network was constructed for the Mawson Lakes neighborhood, with recycled water comprised of a combination of stormwater from the Parafield Airport wetlands and wastewater from the Bolivar Sewage Treatment Plant.

The success of this operation led to the formation of a pioneering business that included nine projects in different locations. Providing non-potable water in a volume equal to $5 \times 10^6$ m$^3$ per year showed that stormwater containing contaminants, when stored underground and under control, can be used for uses such as irrigation of public open spaces and, when chlorinated, can be supplied in pipes (third pipe supplies). Its use for potable purposes depends on the additional use of microfiltration, UV disinfection, and chlorination, but the costs of these additional operations to reach the required safety standards are considered to be lower than the costs of laying double distribution pipes. The total cost of supply (capital and operation), for example, in 2012/13 was AUD 1.57/m$^3$ for non-potable use for irrigation of public spaces and AUD 1.96 to AUD 2.24/m$^3$ for potable use (excluding distribution network costs), therefore cheaper than the usual AUD 3.45/m$^3$ for mains water [25]. The costs of providing non-potable water through a new distribution network, however, are similar to or higher than the costs of distributing water from the existing network.

In 2010, a business unit, Salisbury Water Business Unit, participating in the administrative structure, administered by the SWMB and chaired by an external independent member, was established. The unit manages various water collection and supply schemes for non-potable use, being mainly recycled rainwater and native groundwater. Treated to standards, according to the purposes for which they are intended, it is distributed to parks, reserves, schools, industries, and some residential sectors. It serves over 500 users, among them 31 schools, and generated AUD2.8 million in resources in 2015–2016.

### 3.4.2. Melbourne and Victoria

In Greater Melbourne, 5 million people live in an area of about 10,000 km$^2$ with an average annual rainfall of around 600 mm [26]. There is a fixed annual charge per household, based on property value, which is paid as part of the Waterways and Drainage Charge, regardless of the amount of waterproofed area and the impact it has on drainage systems. In the Australian state of Victoria, the Water Act governs how the Waterways and Drainage Charge should be implemented, but it is unclear how the level of waterproofing may influence the levy.

The theoretical graph in Figure 2 demonstrates how the fixed charge works and allows a reflection on the greater possibilities for incentives for non-sealing that can exist from a variable charging policy [17], which is fairer, collects more resources to support the systems,

and provides incentives to non-sealing and disconnection. These can alleviate the need for extensions and maintenance on stormwater systems and save resources more efficiently.

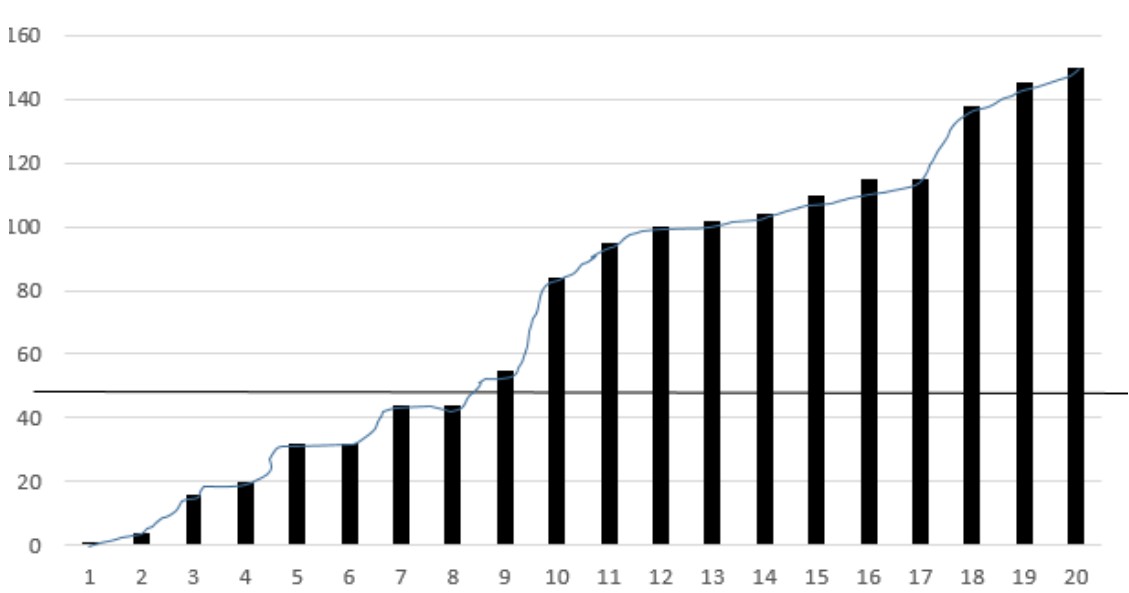

20 different impervious areas (black vertical bars), the impervious fee (curvilinear line) and the flat fee (linear line)

**Figure 2.** Comparison between fixed and variable rates considering the same total storage for 20 dwellings with different degrees of waterproofing [17].

## 4. Results and Discussion

Unlike decades ago, when stormwater management was focused only on urban flooding, there was an evolution that, presents differences and similarities in developed and developing countries. The main issues involve multiple aspects and challenges such as water and environmental quality, aquifer recharge, supply and treatment, urban heat islands, urban well-being, street trees [27] aquatic life, landscaping, and leisure, and flood analysis [28], among many others [29].

Although interrelationships between stormwater and wastewater are known, in several countries, especially where the absolute separator system is adopted, at least officially (as is the case for Brazil), management is still focused on wastewater treatment. This is partly due to the significant sanitation deficit [30], and so stormwater is erroneously considered a low priority, except when flooding events occur [19].

Thus, the collection of stormwater fees and the construction of management structures dedicated to urban stormwater management, with or without private sector participation, has been left behind, especially in developing countries. Initiatives for its implementation are the target of many objections, including judicial ones, as was the case in the municipality of Santo André, in the ABC Paulista, metropolitan region of São Paulo. After a period of evolution and success in the use of charging, started in 1998, a setback occurred, with its suspension in 2012 [31].

Experience, however, including in the U.S., has shown the importance of information and disclosure, particularly when on user demand, with the most used channel of communication with the public being the website [16]. The Web aims at the understanding, involvement, and participation of society and the reduction in objections, proposed mainly by non-residential users. The objection questions are mainly of two types: the legality of the authority responsibility for issuing, implementing, and financing the fees and the legality of the charging mechanism [32].

The organization of management through SWUs considers the institutionalization of the application of fees as an economic support strategy for stormwater management. It is based on four criteria: efficiency, equity, adequacy, and feasibility in the collection and use of revenues. It enables long-term planning of capital and operational investments, brings the potential for change in public behavior, and impacts business and management of municipal investments, but at the same time suffers objections from users, including legal ones.

The tariffs, in general, are based on the operating, maintenance, and, when possible, the capital costs of the systems, distributing them among the users according to criteria that, in most cases, correspond to the waterproofed areas. The criteria seek equity not only through proportionality between tariffs and the contribution to runoff and pollution generated, as established by the user-pays principle, but through different payment capacities, expressed, for example, by property values or consumption of services such as water supply.

The issue of stormwater systems' economic support is still quite controversial, even in places such as the USA and Canada, where there are legal challenges, most of them being refused by the courts. Barriers to their implementation also exist, as can be seen in the Canadian example of the City of Thunder Bay, where the conclusion led to the option, even in the short term, of not adopting the fairest method of distributing the burden of waterproofing.

There are several ways of economic viabilization and funding stormwater management services, but they can be summarised in two, especially with regard to fundraising: the traditional model, based on the general budget (payment by all citizens), and the model based on payment just by the users of the service, known as the stormwater utility.

These two different visions on how to obtain resources to economic support public services are part of a larger dispute involving the role that governments should play in the solution of socio-economic-environmental problems. The traditional vision is opposed to the innovative vision of stormwater utilities, the latter responding in part to the population's desire for more fair tariffs and to reduce the burden of general taxes, as occurred in the USA decades ago, when stormwater utilities were implemented. However, decades after their implementation, the thought that they are hidden taxes disguised under another name still survives.

Existing experiences and those under implementation deserve to be observed, as well as examples extracted from them of what may work and what may not work so well in each context. However, there should be no delay, as this is still a subject that deserves practical experimentation, a kind of learning by doing, given the increasing demands that are coming with the growth in urbanization, rising temperatures, increased urban rainfall and rising sea levels, according to climate change forecasts.

For example, in Brazil, where the focus is still very much on the scarcity of public resources, the introduction of management and funding mechanisms such as SWU, based on the user-pays principle, can work. As in other countries, stormwater utilities in Brasil are welcome, as long as they are applied through policies that encourage not only the economic contribution of users, but that consider forms of management with focus on results. Besides that, they may lead to increasing the involvement and participation, bringing the contribution of all actors to the decisions made, including the design of the calculation methods of charging and the legislation. This may mean opening the way to solutions that lead to economic autonomy of stormwater services management and also for disengagement from the general public budget, decreasing taxes, with gains in responsibility distribution efficiency and a permanent flow of resources to the sector, providing sustained continuity to the actions.

The study shows that the use of the stormwater utilities mechanism is more developed in countries where environmental legislation has been fully implemented. Table 2 shows examples of the countries' main approaches, fee criteria and objectives. From the information in Table 2, although it is not possible to verify uniformity in all aspects, there is

a trend in the approach to control environmental effects (reduce pollution of water bodies) and in the criteria for calculating tariffs (sealed areas).

However, there is no definite trend regarding the form of quantification of the objectives, a fact that can be attributed to the experience and reality in each location. From the perspective of the economic efficiency objective, information on collected revenues compared to measurable cost outcomes of avoided environmental impacts (e.g., volumes of treated effluent) can allow for the ranking of stormwater utilities initiatives. Measured economic parameters also enable the comparison with other alternatives such as the overall budget itself.

This is different in countries where there is no legislation incentives or, as in Brazil, where the law has existed for a long time and is broad, but encounters obstacles to application. Utilities with a low level of institutionalization and administrative and economical disorganization tend to relegate environmental issues, postpone the acquisition of economic support and, eventually, adopt the funding and organization structure.

**Table 2.** Countries, approaches, fee criteria and objectives.

| Country | Main Approach | Practical Fee Measures Criteria | Objective's Evaluation Metrics |
|---|---|---|---|
| EUA | Environment Pollution Conttrol | ERU ($m^2$ of impervious areas); many others (flows, etc) | Pollution and environment statements attendance |
| Germany | Environment and Equity Polluter-Pays Principle | Impervious area and water supply consumption | Impermeable surface reductions, groundwater recharge and treated quantitative vol. reduction |
| Canada | Polluter-Pays Principle | ERU ($m^2$ of impervious area) and "ad valorem" property tax | Impermeable surface reductions |
| Australia | SUWM—Sustainability Urban Water Management | Fix, based on property values | Groundwater recharge measurements; stormwater and reuse of non-potable supply |
| Brasil | Flow Control | $m^2$ of impervious area | Undecided |

## 5. Conclusions

The overview of SWUs presented here provides an update on what has been implemented to ensure the economic sustainability of urban stormwater management systems with the participation not only of the public sector but also of users and private agents, being remunerated as a public service to society as a whole.

The study's contribution comes in the sense of bringing together scattered information and thus allowing the formation of a general picture of the evolution in a certain direction, namely, the economic organization and financial sustainability of urban stormwater drainage and management under a new paradigm, which has been occurring simultaneously in several places around the world.

The perception of this fact as a general trend does more by allowing scholars, researchers, and practitioners to identify it and become aware of what is still missing for its rapid institutionalization, implementation, and experimentation, thus contributing to the evolution and improvement in the sector's actions. This is a small contribution, given what still needs to be done, but with the potential to help transform the reigning mentality, or the business as usual, and in this sense it can be significant.

The institutionalization of charging users for the provision of urban stormwater management services, whether provided by public or private operators, always encounters obstacles, posed by those who believe that they should be compulsorily provided by the public authority and funded by general public budgets, which means the cost is socialized for all the society. Either due to technical reasons, such as methods of quantification of the shares that each one is responsible for (the user-/polluter-pays principle), or to issues of understanding regarding the services to be provided by the state or for various legal

and rights-based reasons (i.e., legal, among others), the fact is that barriers exist to the implementation concept of stormwater utilities.

The reality, however, has shown that in several countries there are feasible ways of charging equitably for the services, relieving public budgets, encouraging the reduction in impervious areas and the disconnection to urban stormwater systems, i.e., saving nature from impacts, taxpayers from unfair costs, and public budgets from unplanned expenses, made to remedy sudden failures after extreme precipitation events, which are increasingly frequent due to the climate.

Based on theoretical knowledge, expressed in various pricing and collection formulas that have been tried in practice in different countries, it is possible to see that economic sustainability, as the economic side of the ongoing paradigm shift in urban stormwater management services already has feasible options and alternatives. Thus, the argument that urban drainage is a public service left "for later" due to a lack of resources or economic organization does not hold. Political will can set in motion policies, institutions, and regulations that, aligned around the objective of solving drainage sector issues, set in motion incentives for economic organisation and financial support.

The novelty is the possibility to make the economic change a viable side of the stormwater paradigm shift, in a win–win manner, with more than economic gains for all actors. There are efficiency gains in terms of environmental, social, institutional, organizational, and political aspects.

This is not all, since society's acceptance of the paradigm shift, through the understanding of the gains that are thus produced, requires an effort of awareness. This is not only motivated by economic–financial gains and reasons, but by others of ideological nature, that is, at the level of ideas and ideals, as is the case for the sustainable development goals. SDG's are present in the paradigm shift, but are not always perceived and require more work from all. A missing economic aspect that is important in ascertaining the speed of the paradigm shift is the quantification of 'green' and 'grey' infrastructure investments made with the revenue raised through SWUs. This is an aspect for study, that is, it is important to know to what extent the mechanism and the collection from users has contributed to the implementation of more infrastructures that favour the paradigm change (e.g., green infrastructures) such as the quantity of street trees and many others leveraging the change. Criteria to measure the achievement of clear objectives make the possibilities of reaching them visible, contribute to adjustments, and can help everything run more quickly towards the change in paradigm.

The study explored the existing publications, information, and data to which it was possible to have access and, by adopting this methodology, it carries with it the limitations arising from it, such as the absence of information that does not exist in the databases studied, or even the form of research used in these bases. Additionally, given the dynamics of the temporal evolution of the experiments, they will continue to occur, often surpassing the ability to become aware of them and analyze them, a fact that is part of the research process.

**Author Contributions:** Conceptualization, C.N. and R.M.; methodology, C.N. and R.M.; validation, C.N. and R.M.; formal analysis, C.N. and R.M.; investigation, C.N.; resources, C.N. and R.M.; data curation, C.N.; writing—original draft preparation, C.N. and R.M.; writing—review and editing, C.N. and R.M.; visualization, C.N. and R.M.; supervision, R.M.; project administration, R.M.; funding acquisition, C.N. and R.M. All authors have read and agreed to the published version of the manuscript.

**Funding:** The authors are grateful for the Foundation for Science and Technology's support through funding UIDB/04625/2020 from the research unit CERIS.

**Institutional Review Board Statement:** Not applicable.

**Informed Consent Statement:** Not applicable.

**Data Availability Statement:** Not applicable.

**Conflicts of Interest:** The authors declare no conflict of interest.

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
