# Peer review of "Stormwater Utilities: A Sustainable Answer to Many Questions"

_sustainability, doi:10.3390/su14106179_

Round 1

Reviewer 1 Report

The paper is based on a review on urban stormwater management. Country-based utilities are assessed in subsections. The subject is very important but the literature is reviewed insufficiently. Some suggestions and comments to the authors are presented below:

1. The abstract is too short. It should be developed

  1. Use passive sentences. Check the sentences started by “we”. See line 234 …
  2. A basic flow chart of the study should be added. Thus, the readers can easily follow procedures.
  3. Check the spaces between words and numbers as “$1 million” in line 223 …
  4. Give last updated examples from literature about urban stormwater management as

Burgan, H.I., Icaga, Y. (2019). Flood analysis using Adaptive Hydraulics (AdH) model in Akarcay Basin. Teknik Dergi, 30(2), 9029-9051. https://doi.org/10.18400/tekderg.416067.

Barbosa, A. E., Fernandes, J. N., David, L. M. (2012). Key issues for sustainable urban stormwater management. Water research, 46(20), 6787-6798. https://doi.org/10.1016/j.watres.2012.05.029.

Berland, A., Shiflett, S. A., Shuster, W. D., Garmestani, A. S., Goddard, H. C., Herrmann, D. L., Hopton, M. E. (2017). The role of trees in urban stormwater management. Landscape and urban planning, 162, 167-177. https://doi.org/10.1016/j.landurbplan.2017.02.017.

  1. There are some crucial and grammatical errors. Check them all.
  2. Conclusions part should be improved. Main conclusions of the study should be explained well.
  3. What is the novelty of the paper? It should be emphasized in the paper.
  4. Use same scheme for variables (italic or not) in the equations and the paragraphs. See Fracres, ERUres
  5. One sentence can’t be a paragraph. See the lines 40-42, 60-64, 76-81, 214-217, 254-256 …
  6. The resolution of figures should be increased. See Figures 1 & 2.
  7. Statistical properties and distribution characteristics of literature data can be given in the paper. Therefore, the trends can be assessed.
  8. There are some tense mistakes. There are present and past tenses in a paragraph. See the first paragraph under Introduction in lines 15-21 …
  9. Check the uppercases as “19th" in lines 26, 47 …

Reviewer 2 Report

The overview is important for discussing what economic instruments to use. The publication has the character of a review – it has not methodology for data processing, tool analysis, comparison or evaluation of efficiency.

Before publishing the text, I recommend:

  1. To publish the text in the review category.
  2. To make a clear comparison of approaches in individual countries: to propose comparison criteria and to prepare a tabular presentation.
  3. It is necessary to discuss whether the examples given are unique (these are regions advanced in stormwater management and economically advanced). What is the situation in other countries? Is there any clear information (e.g. OECD reports)?
  4. How are fee levels defined – e.g. according to cleaning and infrastructure costs or with regard to social suitability? This discussion is important for sustainable stormwater management.

Round 2

Reviewer 1 Report

Some suggestions and comments are partially replied in the first round. Some corrections are still required as suggested below:

  1. Keywords should be ordered A to Z.
  2. One sentence can’t be a paragraph. See the lines 126-132, 176-180, 183-194 …
  3. The authors accept the suggested three papers about urban stormwater management in the first revision. They should be referenced in the paper.
  4. There are some tense mistakes. There are present and past tenses in a paragraph. See the first paragraph under Introduction in lines 23-28 …

Reviewer 2 Report

Recommendations and comments were processed into a manuscript. The text is suitable for publication, but it has the character of a review.
